# Evaluation of Data Augmentation Techniques for Facial Expression Recognition Systems

**Simone Porcu** [1,2], **Alessandro Floris** [1,2,*] **and Luigi Atzori** [1,2]

1. Department of Electrical and Electronic Engineering, University of Cagliari, 09123 Cagliari, Italy; simone.porcu@unica.it (S.P.); l.atzori@ieee.org (L.A.)
2. CNIT, University of Cagliari, 09123 Cagliari, Italy
* Correspondence: alessandro.floris84@unica.it

**Abstract:** Most Facial Expression Recognition (FER) systems rely on machine learning approaches that require large databases for an effective training. As these are not easily available, a good solution is to augment the databases with appropriate data augmentation (DA) techniques, which are typically based on either geometric transformation or oversampling augmentations (e.g., generative adversarial networks (GANs)). However, it is not always easy to understand which DA technique may be more convenient for FER systems because most state-of-the-art experiments use different settings which makes the impact of DA techniques not comparable. To advance in this respect, in this paper, we evaluate and compare the impact of using well-established DA techniques on the emotion recognition accuracy of a FER system based on the well-known VGG16 convolutional neural network (CNN). In particular, we consider both geometric transformations and GAN to increase the amount of training images. We performed cross-database evaluations: training with the "augmented" KDEF database and testing with two different databases (CK+ and ExpW). The best results were obtained combining horizontal reflection, translation and GAN, bringing an accuracy increase of approximately 30%. This outperforms alternative approaches, except for the one technique that could however rely on a quite bigger database.

**Keywords:** facial expression recognition; machine learning; generative adversarial network; data augmentation; convolutional neural network; synthetic image database

## 1. Introduction

Facial Expression Recognition (FER) is a challenging task involving scientists from different research fields, such as psychology, physiology and computer science, whose importance has been growing in the last years due to the vast areas of possible applications, e.g., human–computer interaction, gaming and healthcare. The six basic emotions (i.e., anger, fear, disgust, happiness, surprise and sadness) were identified by Ekman and Freisen as the main emotional expressions that are common among human beings [1]. Although numerous studies have been conducted on FER, it remains one of the hardest tasks for image classification systems due to the following main reasons: (i) significant overlap between basic emotion classes [1]; (ii) differences in cultural manifestation of emotion [2]; and (iii) need of a large amount of training data to avoid overfitting [3]. Moreover, many state-of-the-art FER methods present a misleading high-accuracy because no cross-database evaluations are performed. Indeed, facial features from one subject in two different expressions can be very close in the features space; conversely, facial features from two subjects with the same expression may be very far from each other [4]. For these reasons, cross-database analyses are preferred to improve the validity of FER systems, i.e., training the system with one database and testing with another one.

Large databases are typically needed for training and testing machine learning algorithms, and in particular deep learning algorithms intended for image classification systems, mostly to avoid overfitting. However, although there are some public image-labeled databases widely used to train and test FER systems, such as Karolinska Directed Emotional Faces (KDEF) [5] and Extended Cohn–Kanade (CK+) [6], these may not be large enough to avoid overfitting. Techniques such as data augmentation (DA) are then commonly used to remedy the small dimension and/or class imbalance of these public databases by increasing the amount of training samples. Basically, DA techniques can be grouped into two main types [7]: (i) data warping augmentations generate image data through label-preserving linear transformations, such as geometric (e.g., translation, rotation, scaling) and color transformations [8,9]; and (ii) oversampling augmentations are task-specific or guided-augmentation methods which create synthetic instances given specific labels (e.g., generative adversarial network (GAN)) [10,11]. Although the second can be more effective, their computational complexity is higher.

In this paper, we aim to quantify and compare the effect of using different well-established DA techniques on the emotion recognition accuracy of a FER system based on a deep network. Specifically, we consider geometric transformations (i.e., random rotation, horizontal reflection, vertical reflection, cropping and translation) as well as GAN to generate novel images from the training database to increase the amount of training samples. The synthetic images generated with GAN have also been made public. The augmented training database was the input of the deep network, which in the proposed system is the well-known VGG16 Convolutional Neural Network (CNN) [12]. We based our method on a CNN because it is robust to face location changes and scale variations [3]. As our main objective is to investigate whether the selection of the proper DA technique may compensate the lack of large database for training the FER model, we do not provide a novel CNN but we based on the VGG16. To increase the value of our analysis, we performed cross-database evaluations by training the FER model with the KDEF database and testing with two different databases (CK+ and ExpW). We evaluated and compared the accuracy achieved by using each of the considered DA technique. The combination of the three DA techniques achieving the highest accuracy values, i.e., horizontal reflection, translation and GAN, allowed obtaining about 30% increase in terms of recognition accuracy. Moreover, we compared the values of the sensitivity, specificity, precision and F1 score metrics computed for the single emotion classes when considering the best combination of DA techniques. Finally, we compared the obtained results with those obtained by state-of-the-art studies.

The main contributions of this study are:

- We performed a comparison to investigate which DA technique (including their combination, among geometric and GAN-based) is the more convenient for FER systems. To this, we also performed cross-database evaluation. The reason is that it is not always easy to understand which DA technique should be used to remedy the small dimension of public image-labeled databases because state-of-the-art experiments use different settings which makes the impact of DA techniques not comparable.
- We provide an advancement with respect to the current state-of-the-art as there is limited research in this regard. To the best of the authors' knowledge, the only study proposing a comparison of DA techniques is that of Pitaloka et al. [13], who considered limited DA techniques and smaller databases. Moreover, no cross-database evaluation is performed.
- We consider the specific utilization of GAN as a DA technique for FER systems, which is barely investigated in the literature.

The paper is structured as follows. Section 2 discusses related work. Section 3 presents the proposed FER system. Section 4 discusses the experimental results. Finally, Section 5 concludes the paper.

## 2. Related Work

In this section, we discuss related work proposing FER systems and using DA techniques for FER.

## 2.1. Facial Expression Recognition

Most traditional FER studies consider the combination of face appearance descriptors, which are used to represent face expressions, with deep learning techniques to handle the challenging factors for FER, achieving the state-of-the-art recognition accuracy [3]. Ali et al. [14] used Principal Component Analysis (PCA), Local Binary Pattern (LBP) histogram and Histogram of Oriented Gradient (HOG) for sample image representation, whereas the proposed FER system is based on boosted neural network ensemble (NNE) collections. Experimental results are accomplished on multicultural facial expression datasets (RaFD, JAFFE and TFEID) and the achieved accuracy is comparable to state-of-the-art works. A novel edge-based descriptor, named Local Prominent Directional Pattern (LPDP) was proposed by Makhmudkhujaev et al. [15]. The LPDP considers statistical information of a pixel neighborhood to encode more meaningful and reliable information than the existing descriptors for feature extraction. Extensive experiments on FER on well-known datasets (CK+, MMI, BU-3DFE, ISED, GEMEP-FERA and FACES) demonstrate the better capability of LPDP than other existing descriptors in terms of robustness in extracting various local structures originated by facial expression changes. Shan et al. [16] used LBP as feature extractor and combined different machine learning techniques to recognize facial expressions. The best result are obtained by using LBP and Support Vector Machine (SVM) on the CK+ database. Cross-database validation is also performed training with the CK+ and testing with JAFFE. In Reference [17], a FER system is proposed based on an automatic and more efficient facial decomposition into regions of interest (ROI). These ROIs represent seven facial components involved in expression of emotions (left eyebrow, right eyebrow, left eye, right eye, between eyebrows, nose and mouth) and are extracted using the positions of some landmarks. A multiclass SVM classifier is then used to classify the six basic facial expressions and the neutral state. A cross-database evaluation is also performed training with the KDEF database and testing with the CK+ database. Similarly, Gu et al. [18] divided each image into several local ROIs containing facial components critical for recognizing expressions (i.e., eyebrow corner, mouth corner and wrinkle). Each of these ROI is then subjected to a set of Gabor filters and local classifiers that produce global features representing facial expressions. In-group and cross-database experiments are conducted using CK+ and JAFFE databases. In Reference [19], landmark points are used for the recognition of the six basic facial expressions in images. The proposed technique relies on the observation that the vectors formed by the landmark point coordinates belong to a different manifold for each of the expressions. Extensive experiments are performed on two publicly available datasets (MUG and CK+) yielding very satisfactory expression recognition accuracy. Flávio Altinier Maximiano da Silva [2] investigated the performance of cross-classification using facial expression images from different cultures taken from four databases, i.e., CK+, MUG, BOSPHOROUS and JAFFE. Different combinations of descriptors (HOG, Gabor filters and LBPs) and classifiers (SVM, NNs and k-NNs) are employed for experiments. Experiment results highlight that the most effective combination for in-group classification was formed by the association of HOG filter and SVM. However, when classifying different databases, even with the most effective combination, the accuracy dropped considerably. Hasani and Mahoor [20] proposed a novel Deep Neural Network (DNN) method for FER based on a two-step learning process aimed to predict the labels of each frame while considering the labels of adjacent frames in the sequence. Experimental evaluations performed with three databases (CK+, MMI and FERA) show that the proposed network outperforms the state-of-the-art methods in cross-database tasks. Mollahosseini et al. [21] presented a novel deep neural network architecture for the FER problem, which examines the network's ability to perform cross-database classification. Experiments are conducted on seven well-known facial expression databases (i.e., MultiPIE, MMI, CK+, DISFA, FERA, SFEW and FER2013) obtaining results better than, or comparable to, state-of-the-art methods. Lopes et al. [4] proposed a combination of CNN and image pre-processing steps aimed to reduce the need for a large amount of data by decreasing the variations between images selecting a subset of the features to be learned. The experiments were carried out using three public databases (i.e., CK+, JAFFE and BU-3DFE). Both in-group and cross-database evaluations are performed.

## 2.2. Data Augmentation for FER

Due to the small dimension of public image-labeled databases, DA techniques are commonly used to augment the database's dimension. Geometric DA techniques are used the most, due to their low computational complexity. Simard et al. demonstrated the benefits of applying geometric transformations of training images for DA, such as translations, rotations and skewing [8]. In Reference [9], five DA geometric techniques are included in the FER system: rotation, shearing, zooming, horizontal flip and rescale. Experimental results show that DA boosted the model in terms of accuracy. In Reference [22], two DA techniques are considered in the proposed FER system: random noise and skew. The former adds random noise to the position of the eyes, whereas the latter applies a random skew, i.e., changes the corners of the image to generate a distortion. Extensive cross-database experimentation are conducted using six databases (i.e., CK+, JAFFE, MMI, RaFD, KDEF, BU3DFE and ARFace), and the proposed system outperform state-of-the-art results in most of the cases. In Reference [13], four DA methods are compared as CNN enhancer: resize, face detection and cropping, adding noises and data normalization. The combination of face detection with adding noises for data augmentation boosted the performance of the CNN in terms of accuracy.

Besides geometric transformations, more complex guided-augmentation methods may be used for DA, such as GAN [23]. In Reference [24], a general framework of DA using GANs in feature space for imbalanced classification is proposed. Experiments are conducted on three databases, i.e., SVHN, FER2013 and Amazon Review of Instant Video, showing the significant improvement with feature augmentation of GANs. In Reference [10], a conditional GAN is used to generate images aimed at augmenting the FER2013 dataset. A CNN is used for training the prediction model and the average accuracy obtained 5% increase after adopting the GAN DA technique. In Reference [11], a framework is designed using a CNN model as the classifier and a cycle-consistent adversarial networks (CycleGAN) as the generator for data augmentation. Experiments on three databases (FER2013, JAFFE and SFEW) show that the proposed GAN-based DA technique can obtain 5–10% increase in the accuracy. In Reference [25], a FER method based on Contextual GAN is proposed, which uses a contextual loss function to enhance the facial expression image and a reconstruction loss function to maintain the identity information of the subject in the expression image. Experimental results on the augmented CK+ and KDEF databases show that this method improves the recognition accuracy by about 5%. However, none of these studies performs cross-database evaluation.

## 3. The Proposed FER System

The framework of the proposed FER system is shown in Figure 1.

The dashed line separates the training phase from the testing phase. The operations performed during the training phase are the following. Firstly, a face detection operation is performed to detect and select only the face information from the training images, which are acquired from the selected training database. The emotion labels associated to each training image are used by the CNN to train the Emotion Prediction model. Different DA techniques are implemented to augment the training database, namely: random rotation (RR), horizontal reflection (HR), vertical reflection (VR), cropping (CR), translation (TR) and generative adversarial network (GAN). These techniques have the objective to increase the size of the training database for the model training process by producing novel but different versions of the original images. The augmented images are resized to be compatible with the CNN accepted image size. The CNN is trained with these images and its output is the Emotion Prediction model, which is able to process a face image and predict one of the six basic human emotions (anger, sadness, surprise, happiness, disgust and fear) or the neutral emotion. The experiment design aims to identify the impact of each DA technique on the performance of the Emotion Prediction model, which are computed using the testing database, which does not contain any image used for training the model so as to perform cross-database evaluation.

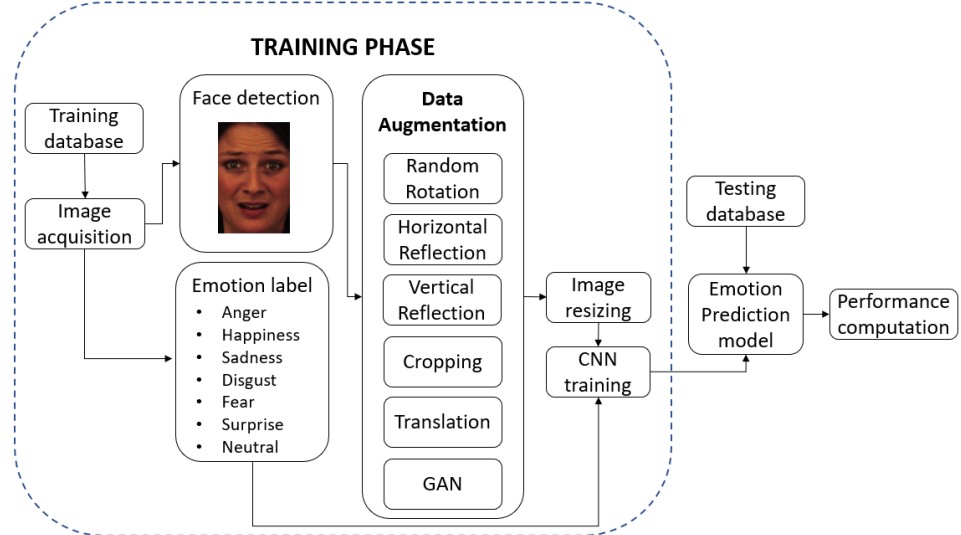

**Figure 1.** The proposed FER system. The dashed line separates the training phase from the testing phase.

In the following sections, we provide the details of the major elements of the proposed FER system.

### 3.1. Face Detection

The face detection operation allows for selecting only the face information from the input image by removing all unnecessary data for emotion recognition. Firstly, the RGB input image is converted into a gray-scale image. Then, for the face detection operation, we used the S$^3$FD (Single Shot Scale-invariant Face Detector) [26], which solves the problem of anchor-based detection methods whose performance decrease rapidly as the faces becoming smaller.

### 3.2. Geometric DA Techniques

- Random Rotation: The RR technique applies a random rotation between $-90°$ and $+90°$ to the image.
- Horizontal reflection: HR, also known as horizontal flip, is a DA technique that creates a mirrored image from the original one along the vertical direction.
- Vertical reflection: VR, also known as vertical flip, is a DA technique that creates a mirrored image from the original one along the horizontal direction. A VR is equivalent to rotating an image by $180°$ and performing a HR.
- Translation: The TR DA technique performs a random moving of the original image along the horizontal or vertical direction (or both). Padding zeros are added to the image sides.
- Cropping: The CR DA technique randomly samples a section from the original image. The cropped image size is large enough to contain a relevant part of the face.

### 3.3. Generative Adversarial Network

GAN is a framework including two deep networks, one (the generative) pitted against the other (the discriminative). This approach allows the neural network to create new data with the same distribution of training data. The generator attempts to produce a realistic image to fool the discriminator, which tries to distinguish whether this image is from the training set or the generated set [11]. In the proposed FER system, we used the GAN implemented for the DeepFake autoencoder architecture of the FaceSwap project (https://github.com/deepfakes/faceswap). In Figure 2, we show some novel synthetic images generated with GAN. The face images from the KDEF database are

used as the base to create novel synthetic images using the facial features of two images (i.e., Candie Kung and Cristina Saralegui) selected from the YouTube-Faces database [27]. It can be seen as the novel images differ between each other, in particular with respect to the eyes, nose and mouth, whose characteristics are taken from the Candie and Cristine images. We created a database with these novel synthetic images that we have made public (https://mclab.diee.unica.it/?p=272).

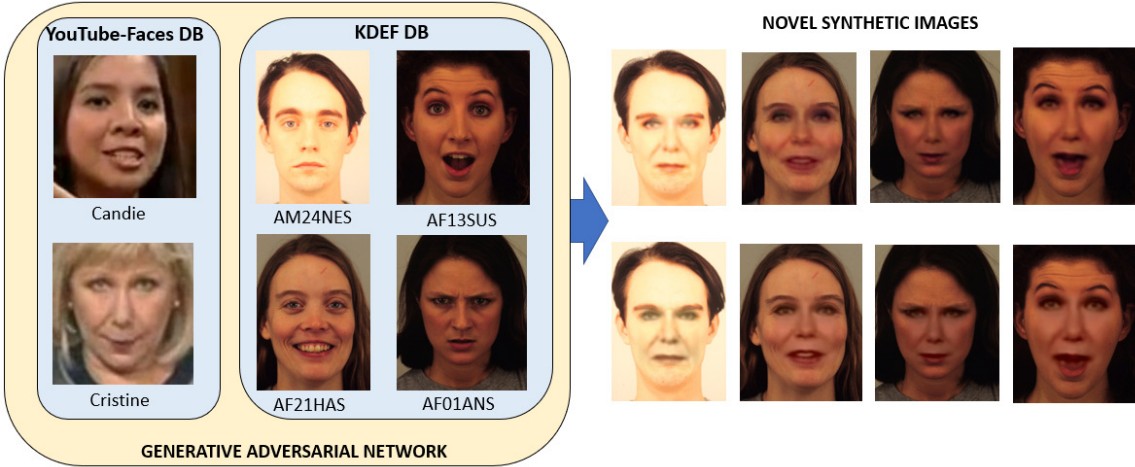

**Figure 2.** Examples of novel synthetic images generated with the GAN.

### 3.4. Convolutional Neural Network

The CNN considered for the proposed FER model is the VGG16, a popular CNN proposed by Simonyan and Zisserman, which competed in the ImageNet Large Scale Visual Recognition Challenge achieving a top-5 accuracy of 92.7% [12]. The VGG16 has made improvements over the AlexNet CNN by replacing large kernel-sized filters (11 and 5 in the first and second convolutional layers, respectively) with multiple $3 \times 3$ kernel-sized filters one after another. The architecture of the VGG16 is shown in Figure 3.

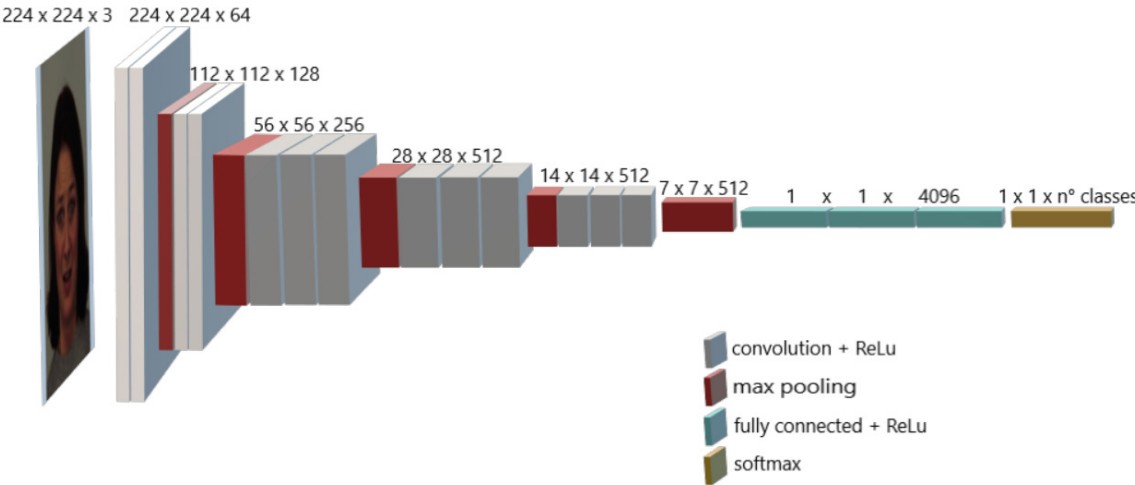

**Figure 3.** Architecture of the VGG16 CNN.

## 4. Experimental Results

The experiments were performed using three publicly available databases in the FER research field: KDEF [5], CK+ [6] and ExpW [28]. The KDEF database is a set of 4900 pictures of human facial expressions with associated emotion label. It is composed of 70 individuals, each displaying the six basic emotions plus the neutral emotion. Each emotion was photographed (twice) from five

different angles, but for our experiments we only used the frontal poses, for a total of 490 pictures. The CK+ database includes 593 video sequences recorded from 123 subjects. Among these videos, 327 sequences from 118 subjects are labeled with one of the six basic emotions plus the neutral emotion. The ExpW database contains 91,793 faces labeled with the six basic emotions plus the neutral emotion. This database contains a quantity of images much larger than the others databases and with more diverse face variations.

A cross-database evaluation process was performed by training the proposed FER model with the KDEF database augmented with the considered DA techniques with the following setting: (1) 490 images created with RR; (2) 490 images created with HR; (3) 490 images created with VR; (4) 490 images created randomly translating 20% of each image (optimal setting found empirically); (5) 490 images created randomly cropping 50% of each image (optimal setting found empirically); and (6) 980 images generated with the GAN (70 individuals from KDEF database × 7 emotions × 2 subjects from YouTube-Faces database). The size of the training database was 980 images when using RR, HR, VR, TR and CR DA techniques and 1470 images when using GAN.

The DA techniques as well as the CNN were implemented with PyTorch [29] and the related libraries. The experiments were conducted with a Microsoft Windows 10 machine with the NVIDIA CUDA Framework 9.0 and the cuDNN library installed. All the experiments were carried out using an Intel Core i9-7980XE with 64 GB of RAM and two Nvidia GeForce GTX 1080 Ti graphic cards with 11 GB of GPU memory each.

The testing was performed with CK+ (first experiment) and ExpW databases (second experiment). Table 1 summarizes the values of the overall emotion recognition accuracy achieved considering different DA techniques. Specifically, we computed the macro-accuracy; however, we refer to it in the rest of the paper only as accuracy. Figures 4 and 5 show a comparison of the sensitivity, specificity, precision and F1 score metrics computed for the single emotion classes when considering the best combination of DA techniques. The sensitivity and specificity highlight, respectively, the number of correct positive and correct negative predictions (True Positive Rate and True Negative Rate), whereas the precision identifies the frequency of correct predictions for the actual positive instances. Finally, the F1 score allows analyzing the trade-off between sensitivity and precision.

**Table 1.** Comparison of the overall emotion recognition accuracy achieved by the proposed FER system considering different DA techniques. Training database, KDEF; testing databases, CK+ and ExpW.

| Data Augmentation | Testing Database: CK+ | Testing Database: ExpW |
|---|---|---|
| | Accuracy | Accuracy |
| No DA | 53% | 15.7% |
| CR | 50% | - |
| RR | 53% | - |
| VR | 53% | - |
| HR | 57% | - |
| TR | 61% | - |
| GAN | 75% | - |
| HR and TR | 63% | 20% |
| GAN and HR and TR | 83.3% | 40.4% |

### 4.1. First Experiment: Testing with the CK+ Database

In the first experiment (testing with the CK+ database), when no DA techniques were used, i.e., the FER system was trained with the original dataset, the achieved overall accuracy is 53%. CR is the only DA technique that causes a reduction of the emotion recognition accuracy (50%) while the RR and VR DA techniques do not bring any enhancement with the system achieving the same accuracy with respect to the no DA case (53%). The HR and TR DA techniques allow the FER system to slightly improve the performance by achieving accuracies of 57% and 61%, respectively. The single DA technique that permits the FER system to reach the greatest accuracy, i.e., 75%, is GAN.

The combination of the two geometric DA techniques singularly achieving the greatest accuracy, i.e., HR and TR, brings a 10% increase in accuracy with respect to the no DA case (63% vs. 53%). However, the combination of these two techniques with the GAN allows achieving the greatest overall emotion recognition accuracy, i.e., 83.3%, with a 30.3% increase when compared with the no DA case. We want to specify that the GAN generated images were not augmented with the geometric transformation, but just added to the set of geometric augmented normal images. Therefore, the size of the training database when augmenting with the GAN, HR and TR combination was 2450.

Furthermore, in Figure 4, we compare the sensitivity, specificity, precision and F1 score metrics computed for the single emotion classes when considering the best combination of DA techniques, i.e., GAN, HR and TR. From this comparison, it can be seen that sadness and disgust are the only emotions that are always 100% recognized. The happiness and neutral emotions are also recognized with great sensitivity (100%) and specificity (greater than 90%) but lower precision and F1 score, which highlight a greater rate of false positives for these emotions. Finally, anger, surprise and fear are the emotions achieving the worst performance. Indeed, although specificity is quite high (greater than 90%) for all emotions, the sensitivity is, respectively, 50%, 63% and 78%, highlighting a significant rate of false negatives. This can be justified by the similar face expressions manifested by the people when considering these three emotions.

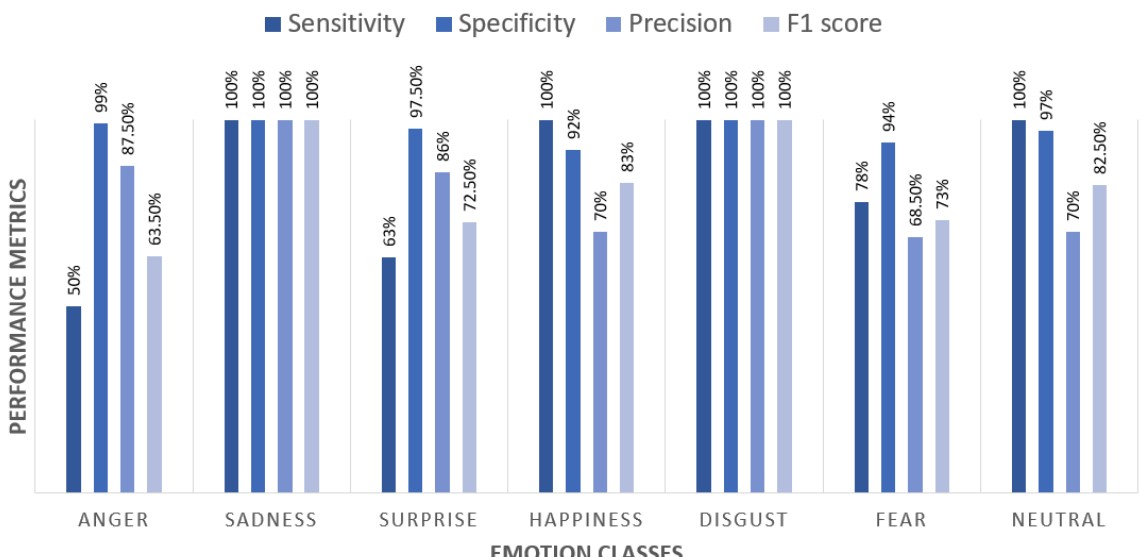

**Figure 4.** Comparison of the sensitivity, specificity, precision and F1 score metrics computed for the single emotion classes when considering the best combination of DA techniques, i.e., GAN, HR and TR. Training database, KDEF; testing database, CK+.

## 4.2. Second Experiment: Testing with the ExpW Database

In the second experiment, the system was tested with the ExpW database, which contains a quantity of images much larger than the training database (KDEF) and with a greater number of different people with more diverse face variations. Indeed, this database is composed of images taken "in the wild", i.e., emotions that people are naturally manifesting while doing some action. This is quite different from the images included in the KDEF and CK+ databases, which are specifically taken when people are posing. We computed the accuracy only for the case of no DA and for the combinations of DA techniques that achieved higher accuracy in the first experiment (see Table 1). It can be seen that the accuracy achieved by training only with the original dataset is really low, i.e., 15.7%. The combination of HR with TR techniques allows increasing the accuracy up to 20%. However, also in this case, the greatest accuracy is obtained using a combination of GAN with HR and TR for augmenting the training dataset. The achieved accuracy is 40.4%, i.e., 24.7% higher than the case of

no DA and comparable to the 30.2% increase obtained in the first experiment. This confirms that this combination of DA techniques is effective to improve the emotion recognition accuracy of FER systems, even when performing cross-database testing comparing posing emotions with "in the wild" emotions.

Moreover, in Figure 5, we compare the sensitivity, specificity, precision and F1 score metrics computed for the single emotion classes when considering the best combination of DA techniques, i.e., GAN, HR and TR. Firstly, it can be seen as in general the achieved results are significantly lower than those achieved in the first experiment. However, this was expected because of the more relevant difference between the images included in the training and testing databases. In this case, the neutral emotion is the best recognized emotion with all metrics higher than 75%. The surprise and happiness emotions achieved similar good results, with sensitivity around 60%, specificity around 90% and precision around 50%. On the other hand, the anger, sadness, disgust and fear emotions achieved very low results with sensitivity and specificity around 20% and 25%, respectively. We can conclude that these emotions are the most difficult to be recognized by the proposed FER system when considering images taken "in the wild". The reason can be that these emotions are too similar between each other when the context is "wild" and people are not posing, so that the FER system goes wrong with the classification bringing lower performance.

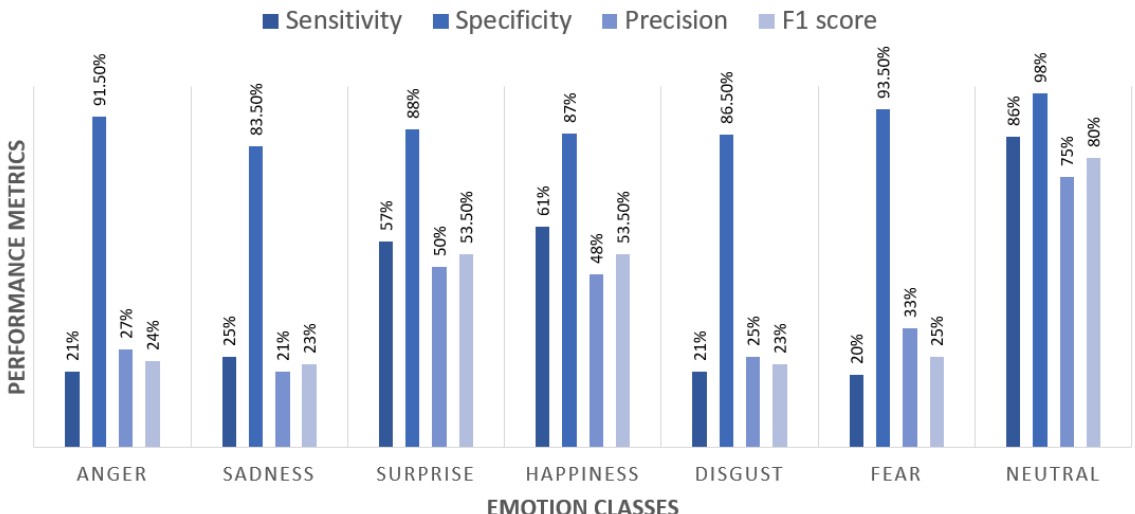

**Figure 5.** Comparison of the sensitivity, specificity, precision and F1 score metrics computed for the single emotion classes when considering the best combination of DA techniques, i.e., GAN, HR and TR. Training database, KDEF; testing database, ExpW.

*4.3. Comparison with the State-of-the-Art*

Finally, in Table 2, we compare the greatest accuracy achieved by the proposed FER system with those achieved by state-of-the-art cross-database experiments conducted for FER systems and tested on the CK+ database. It can be seen that the approach proposed by Zavarez et al. [22] is the only one that outperforms our proposed approach. However, they trained their model using a bigger dataset composed by six databases (more than 6200 images). Furthermore, it must also be noted that our results are achieved by using only two images to generate novel synthetic training images with the GAN. By augmenting the training database with additional images, the performance would likely improve, which will be the objective for future experiments. Further details regarding the state-of-the-art studies considered for the comparison are provided in Section 2.

**Table 2.** Comparison among state-of-the-art cross-database experiments tested on the CK+ database.

| Method | Training Database | Accuracy |
|---|---|---|
| Proposed | KDEF | 83.30% |
| da Silva et al. [2] | MUG | 45.60% |
| da Silva et al. [2] | JAFFE | 48.20% |
| da Silva et al. [2] | BOSPHOROUS | 57.60% |
| Lekdioui et al. [17] | KDEF | 78.85% |
| Gu et al. [18] | JAFFE | 54.05% |
| Hasani et al. [20] | MMI+FERA | 73.91% |
| Mollahosseini et al. [21] | 6 databases | 64.20% |
| Zavarez et al. [22] | 6 databases | 88.58% |

## 5. Conclusions

We experimented on the use of GAN-based DA, combined with other geometric DA techniques, for FER purposes. The "augmented" KDEF database was trained using the well-known VGG16 CNN neural network and the CK+ and ExpW databases were used for testing.

The results demonstrate that geometric image transformations, such as HR and TR, provide limited performance improvements; differently, the adoption of GAN enriches the training database with novel and useful images that allow for quite significant accuracy improvements, up to 30% with respect to the case where DA is not used. These results confirm that FER systems need a huge amount of data for model training and foster the utilization of GAN to augment the training database as a valid alternative to the lack of huge training database. The drawback of GAN is the relevant computational complexity that introduces significant times for the generation of the synthetic images. Specifically, in our experiments, three days were needed to reach a loss value of 0.02 for training the GAN network, from which we obtained the augmented dataset that permitted to reach 83.30% accuracy. This result highlights the relevance of training data for emotion recognition tasks and the importance of considering the correct DA technique for augmenting the training dataset when large datasets are not available. A possible solution to reduce the time needed to train the GAN network may be to generate a lower number of synthetic images with the GAN and combine this technique with less complex geometric DA techniques.

**Author Contributions:** Conceptualization, A.F.; Formal analysis, S.P.; Funding acquisition, L.A.; Investigation, S.P. and A.F.; Methodology, S.P.; Project administration, L.A.; Supervision, L.A.; Writing—original draft, S.P. and A.F.; Writing—review and editing, A.F. and L.A. All authors have read and agreed to the published version of the manuscript.

**Funding:** This work was supported in part by Italian Ministry of University and Research (MIUR), within the Smart Cities framework (Project: Netergit, ID: PON04a200490).

**Conflicts of Interest:** The authors declare no conflict of interest.

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
