# Peer review of "Evaluation of Data Augmentation Techniques for Facial Expression Recognition Systems"

_electronics, doi:10.3390/electronics9111892_

Round 1
Reviewer 1 Report
The paper proposes a method to best use data augmentation in facial expression recognition. The authors compare the performance of their method with several state-of-the-art ones. The result shows that the proposed method performs better.
The paper is well organized and easy to follow. I do not have much comment on the technical details. A minor comment is the overuse of acronyms. CNN might be known by many experts but should be written in the full form first. Also, I personally don't think it is necessary to with DB as database.
Author Response
We thank the Reviewer for his comments. We have written CNN in the full form the first time it appears in the Abstract and in the Introduction. Also, we are not using acronym for database now.
Reviewer 2 Report
In this paper, the authors presented the experiments for evaluate the benefit of data augmentation in facial express recognition. I am afraid this paper can not be published in the current form. 1. The contribution of this paper is quite limited. I suggest the author clarify the contribution of this paper. 2. More visual comparison results should be provided to demonstrate the benefit of DA.Author Response
Comment 1:
The contribution of this paper is quite limited. I suggest the author clarify the contribution of this paper.
Response to the Reviewer:
We added the following text in the Introduction to better highlight the contribution of the proposed study.
“The main contributions of this study are:
- We perform a comparison to investigate which DA technique (and their combination, among geometric and GAN-based) is the more convenient for FER systems. To this, we also perform cross-database evaluation. The reason is that it is not always easy to understand which DA technique should be used to remedy the small dimension of public image-labeled databases because state-of-the-art experiments use different settings which makes the impact of DA techniques not comparable.
- We provide an advancement with respect to the current state-of-the-art as there is limited research in this regard. To the best of the authors' knowledge, the only study proposing a comparison of DA techniques is [13], which considered limited DA techniques and smaller databases. Moreover, no cross-database evaluation is performed.
- We consider the specific utilization of GAN as a DA technique for FER systems, which is barely investigated in the literature.”
Comment 2:
More visual comparison results should be provided to demonstrate the benefit of DA.
Response to the Reviewer:
In the revised article, we are providing additional results and comments. In particular, in Section 4, we have added Figures 4 and 5, which provide a visual comparison of the sensitivity, specificity, precision and F1 score metrics computed for the single emotion classes when considering the best combination of DA techniques. Figures 4 and 5 provide performance comparison when testing the proposed FER model with the CK+ and ExpW database, respectively. Moreover, we provide additional comments regarding these comparisons in the related Sections 4.1 and 4.2.
Reviewer 3 Report
1. In the method section within the "3.3.Convolutional Neural Network", the configuration of the designed neural network model configuration such as layers, and nodes needs to be represent as a figure in detail.
2. In the result part, in particulary, although the highest result was described, buth the lowest case of the accuracy can be explained why the "Anger" emotion has a 20.7 % accuracy with the Testing DB: ExpW.
Author Response
Comment 1:
In the method section within the "3.3.Convolutional Neural Network", the configuration of the designed neural network model configuration such as layers, and nodes needs to be represent as a figure in detail.
Response to the Reviewer:
We added in Fig. 3 the architecture of the considered VGG16 CNN.
Comment 2:
In the result part, in particulary, although the highest result was described, buth the lowest case of the accuracy can be explained why the "Anger" emotion has a 20.7 % accuracy with the Testing DB: ExpW.
Response to the Reviewer:
In the revised article, we are providing additional results and comments. In particular, in Section 4 we have added Figures 4 and 5, which provide a visual comparison of the sensitivity, specificity, precision and F1 score metrics computed for the single emotion classes when considering the best combination of DA techniques. Figures 4 and 5 provide performance comparison when testing the proposed FER model with the CK+ and ExpW database, respectively. Moreover, we provide additional comments regarding these comparisons in the related Sections 4.1 and 4.2, where we also explain the possible reasons regarding the emotions recognized with the lowest performance. One of the main reasons, when testing with the ExpW database, is that the images contained in this database are taken ‘in the wild’, i.e., they show emotions naturally manifested by people while doing some action. This is opposite from the images included in the KDEF and CK+ databases, where people are posing to show a specific emotion.
Round 2
Reviewer 2 Report
The authors made a revision that solves my concerns.